# Bioaccessible Raspberry Extracts Enriched in Ellagitannins and Ellagic Acid Derivatives Have Anti-Neuroinflammatory Properties

**DOI:** 10.3390/antiox9100970

**Published:** 2020-10-10

**Authors:** Gonçalo Garcia, Teresa Faria Pais, Paula Pinto, Gary Dobson, Gordon J. McDougall, Derek Stewart, Cláudia Nunes Santos

**Affiliations:** 1Instituto de Biologia Experimental e Tecnológica (iBET), apartado 12, 2781-901 Oeiras, Portugal; ggarcia@campus.ul.pt (G.G.); mtpais@igc.gulbenkian.pt (T.F.P.); 2ITQB, Instituto de Tecnologia Química e Biológica António Xavier, Universidade Nova de Lisboa, Av. da República, 2780-157 Oeiras, Portugal; paula.pn2@gmail.com; 3Research Institute for Medicines (iMed.ULisboa), Faculty of Pharmacy, Universidade de Lisboa, 1649-003 Lisboa, Portugal; 4Instituto Gulbenkian de Ciência (IGC), Rua Quinta Grande, 2780-156 Oeiras, Portugal; 5Instituto Politécnico de Santarém, Escola Superior Agrária, Qta do Galinheiro, 2001-904 Santarém, Portugal; 6Life Quality Research Centre (CIEQV), IPSantarém/IPLeiria, 2040-413 Rio Maior, Portugal; 7Plant Biochemistry and Food Quality Group, Environmental and Biochemical Science, The James Hutton Institute, Dundee DD2 5DA, Scotland, UK; Gary.Dobson@hutton.ac.uk (G.D.); gordon.mcdougall@hutton.ac.uk (G.J.M.); derek.Stewart@hutton.ac.uk (D.S.); 8School of Engineering and Physical Sciences, Institute of Mechanical, Process and Energy Engineering, Heriot-Watt University, Edinburg EH14 4AS, Scotland, UK; 9CEDOC, Chronic Diseases Research Centre, NOVA Medical School//Faculdade de Ciências Médicas, Universidade NOVA de Lisboa, Campo dos Mártires da Pátria, 130, 1169-056 Lisboa, Portugal

**Keywords:** polyphenol metabolites, gastro-intestinal digestion, neuroinflammation, microglia, NF-kappaB

## Abstract

Chronic neuroinflammation associated with neurodegenerative disorders has been reported to be prevented by dietary components. Particularly, dietary (poly)phenols have been identified as having anti-inflammatory and neuroprotective actions, and their ingestion is considered a major preventive factor for such disorders. To assess the relation between (poly)phenol classes and their bioactivity, we used five different raspberry genotypes, which were markedly different in their (poly)phenol profiles within a similar matrix. In addition, gastro-intestinal bio-accessible fractions were produced, which simulate the (poly)phenol metabolites that may be absorbed after digestion, and evaluated for anti-inflammatory potential using LPS-stimulated microglia. Interestingly, the fraction from genotype 2J19 enriched in ellagitannins, their degradation products and ellagic acid, attenuated pro-inflammatory markers and mediators CD40, NO, TNF-α, and intracellular superoxide via NF-κB, MAPK and NFAT pathways. Importantly, it also increased the release of the anti-inflammatory cytokine IL-10. These effects contrasted with fractions richer in anthocyanins, suggesting that ellagitannins and its derivatives are major anti-inflammatory (poly)phenols and promising compounds to alleviate neuroinflammation

## 1. Introduction

Microglia are the resident innate immune cells in the central nervous system (CNS). Normally, microglia exist in a homeostatic “resting” state, which includes active immunosurveillance of any danger signal. However, microglia become activated upon exposure to signals from the neuronal secretome [1] or from invading pathogens [2]. Classical pro-inflammatory activation is commonly associated with neuroinflammation observed in neurodegenerative diseases [3] and is characterized by dramatic changes in canonical inflammatory-signaling cascades that induce upregulation of many cell surface receptors and cytokines, as tumor necrosis factor-alpha (TNF-α) [4]. Although not disease-exclusive, these pro-inflammatory microglia are more abundant in the brains of patients suffering from neurodegenerative diseases, such as Alzheimer’s (AD) or Parkinson’s (PD) diseases [5,6]. In most cases, due to neuronal dysfunction, microglia are unable to resolve pro-inflammatory cascades and, as a consequence, excessive neuroinflammation exacerbates neurodegeneration [7].

As a critical contributor to both health and disease, diet is an essential modulator of both inflammation and neuroinflammation. In fact, some dietary components, such as (poly)phenols, which can be found in many fruits and vegetables, as well as in nuts and whole grains, have been described as capable of attenuating pro-inflammatory processes associated with chronic diseases [8]. Among them, raspberries represent a good natural source of (poly)phenols with anti-inflammatory properties, such as ellagic acid, flavanols and phenolic acids [9,10], being also rich in other healthy compounds, such as *β*-sitosterol and vitamins C, E and folate. (Poly)phenol-enriched extracts from raspberry have been demonstrated to exhibit remarkable in vivo anti-inflammatory properties [11,12], as well as motor function improvement in aged rats [13]. Other studies also highlighted that a flavonoid-rich diet inhibits inflammatory aggravation in chronic disorders such as AD [14] or rheumatoid arthritis [12]. Although the mechanisms behind such neuroprotective activity are multiple and complex, studies in BV-2 microglia have revealed that raspberry (poly)phenols not only inhibit amyloid-β fibrillation, but also reduce H₂O₂-induced reactive oxygen species (ROS), lipopolysaccharide (LPS)-induced nitric oxide (NO) release, and caspase-3/7 activity [15]. In addition, neolignans found in raspberry have been reported to protect SH-SY5Y neuronal cells against H₂O₂-induced oxidative injury [16].

Unfortunately, many of the beneficial effects demonstrated in vitro by (poly)phenols as anthocyanins have a limited in vivo transferability. This is mostly because the original (poly)phenol molecules are highly susceptible to digestion, absorption and metabolism, and produce (poly)phenol metabolites in vivo that have different chemical and biological properties [10,17]. For example, although anthocyanins have long been recognized for their anti-oxidative and potential anti-inflammatory activities [18], their metabolites have demonstrated lower bioactivity after intestinal digestion [19]. Conversely, many other (poly)phenol metabolites exhibit higher anti-inflammatory potential, when compared to their precursor molecules [20].

In our previous study, we showed that a gastro-intestinal bio-accessible (GIB) fraction obtained from a commercial raspberry variety (*Himbo-Top*) was able to exhibit significant anti-neuroinflammatory and neuroprotective effects, when tested at human physiologically relevant doses [10]. However, we were not able to deduce which classes of compounds present in the GIB fraction were responsible for the observed protective effects. Therefore, in the present study we selected five raspberry genotypes that provided GIB fractions with diverse phytochemical compositions and assessed their neuroprotective effects in a neuroinflammation model based on LPS-stimulated microglia. One of the GIB fractions (from raspberry accession 2J19) revealed remarkable anti-inflammatory potential. Curiously, this accession is characterized by a distinct (poly)phenolic fingerprint compared to the other raspberries, with markedly higher levels of ellagitannins and ellagic acid derivatives and almost no anthocyanins. Therefore, we explored the mechanisms involved in such an anti-inflammatory response through examination of upstream inflammatory pathways MAPK, NFATc1 and NF-κB. Our findings provide new insights that can be useful for the breeding of raspberries with higher levels of these potentially neuroprotective compounds that can now be isolated or engineered to be explored as therapeutic drugs.

## 2. Materials and Methods

### 2.1. Plant Material and Characterization of Phenolic Diversity

Fruit from raspberry cultivars Glen Cally; Glen Doll; Glen Ericht; Glen Fyne; Tulameen; and genotypes (breeding lines) 2J19, 00123A7, 0019E2, 0304F6, 0435D-3, 0460F-5, 0485K-1, 0534RB-1, 9455F-2, and 9911C-1 from the Hutton breeding program were harvested in summer 2012 when fully ripe. The berries were frozen, freeze dried then ground to powder using a mortar and pestle in the presence of liquid nitrogen.

Freeze-dried raspberry powders (500 ± 2 mg) were extracted in 50% acetonitrile as described previously [21]. Samples were assayed for total phenol content using the Folin method [22]. Aliquots (0.5 mL) were dried by centrifugal evaporation (Speed Vac at 45 °C for 3 h) and stored at −20 °C.

### 2.2. In Vitro Digestion of Berry Powders

Selected raspberry samples were subjected to an in vitro digestion (IVD) procedure, as previously described [22,23], but only the gastro-intestinal fractions assumed to be bio-accessible to serum (termed GIB) were used. After IVD, the GIB fractions were subjected to solid phase extraction on C18 units as described previously [24] and then dried by centrifugal evaporation.

### 2.3. Liquid Chromatography-Photodiode Array-Mass Spectrometry (LC-PDA-MS)

The dried raspberry extracts were resuspended in 30% aqueous acetonitrile containing 0.5% formic acid (475 µL), and morin (25 µL of 0.25 mg mL^−1^ in methanol;) was added as an internal standard. Dried GIB fractions were dissolved in ultra-pure water (0.5 mL). Samples were subjected to LC-PDA-MS on a Thermo Fisher Scientific (Hemel Hempstead, UK) LTQ Orbitrap XL ion trap mass spectrometer system as described previously [21]. Anthocyanins were identified in positive mode from exact masses derived from [M+H]+ ions and MS2 data, and all other compounds were similarly identified in negative mode. Compounds were quantified using peak areas and processing methods created in XcaliburTM software, version 2.0 (Thermo Fisher Scientific, Hemel Hempstead, UK) against calibration curves of a standard anthocyanin (cyanidin-3-*O*-sophoroside) and flavonol (quercetin-3-*O*-glucuronide; ExtraSynthase Ltd., Genay, France).

### 2.4. Cell Culture Maintenance, Incubation with GIB Fractions and LPS Stimulation

The N9 murine microglial cell line was maintained in EMEM (Eagle Minimum Essential Media) supplemented with 10% fetal bovine serum (FBS) (Gibco), 1% (*v*/*v*) non-essential amino acids (NEAA) (Sigma–Aldrich^®^—Poole, Dorset, UK), L-glutamine (200 mM), and cultured at 37 °C under humidified atmosphere with 5% CO_2_ (*v*/*v*), accordingly with previous studies [10,25]. For assay, N9 murine microglial cells were suspended and cultured onto 6-well plates and incubated overnight before pre-incubation with each GIB fraction at 1 µg GAE·mL^−1^, for up to 6h. Subsequently, cells were stimulated with LPS (from *E. coli* 055:B5, Sigma–Aldrich) at 300 ng·mL^−1^ during 24 h for NO/cytokine release, or intervals between 15–60 min for determination of transcription factor activation by Western blot and immunocytochemistry.

### 2.5. Cytotoxicity Determination

Cytotoxicity of raspberry GIB fractions in the N9 murine microglial cell line was tested using the cell viability assay (CellTiter-Blue Cell Viability Assay, Promega, Madison, WI, USA), as previously described [22]. After 24 h, four concentrations (0.25, 0.5, 1, and 2 µg GAE mL^−1^) of each GIB fraction were incubated with cells in 0.5% FBS media. After 21 h, cell viability was assessed using a Synergy HT microplate reader (Biotek, Winusky, VT, USA) and viability values normalized to control.

### 2.6. Griess Reaction and ELISA

Quantification of NO accumulation in cell media was performed using the Griess Reagent (Sigma–Aldrich^®^—Poole, Dorset, UK), as described before [10]. For analysis, 100 µL of cell culture medium of each condition were used and the manufacturer’s instructions followed. The absorbance (Abs) was read at 540 nm in a Synergy HT microplate reader (Biotek). For ELISA, cell media was collected from the plate and immediately frozen at −80 °C until analysis. TNF-α release was assayed by sandwich ELISA according to the manufacturer’s instructions (PeproTech^®^, Princeton Business Park, Rocky Hill, NJ, USA) as previously performed [10]. The Abs_405_ was measured using a Synergy HT microplate reader (Biotek) at room temperature (RT). Adherent cells were lysed with RIPA for protein quantification and Griess/ELISA results were normalized for total protein.

### 2.7. Flow Cytometry

Culture media was discarded and cells were mechanically detached using 1 mL PBS, spun-down and incubated for 30 min at 4 °C with mouse anti-FcγR (same as CD-16/32, E-Biosciences) diluted in FACS buffer (PBS containing 2% fetal calf serum and 0.01% NaN_3_), to block unspecific Fc-mediated reactions [25]. Then, cells were spun-down, washed and double-stained with 5 µg·mL^−1^ mouse anti-CD40-FITC conjugated antibody (clone 3/23, BD Biosciences); and with 5 µg·mL^−1^ dihydroethidium (DHE) probe (Invitrogen™, Carlsbad, CA, USA) as an indicator of intracellular superoxide. Fluorescence was acquired in a CUBE 6 cytometer (Sysmex-Partec, Arndtstraße, Goerlitz, Germany). Post-acquisition analysis was performed with FSC express 4 flow research edition software.

### 2.8. Protein Extraction, Quantification and Western Blotting

Cells were washed with PBS and lysed using RIPA buffer supplemented with cocktail protease inhibitors (AppliChem Inc—Mary Avenue, MO, USA), PhosSTOP^TM^ (Roche^®^—Basel, Switzerland) and 0.4% (*v*/*v*) and DNAse (Roche). After 15 min of incubation at room temperature, lysates were scrapped, transferred into microtubes, centrifuged (for 10 min; 4 °C; 8000× *g*), snap frozen, and stored at −80 °C until protein determination by Bradford protein assay. For Western Blot, 40 μg of total protein were separated on 12% acrylamide (*w*/*v*) SDS-PAGE gels. After transfer, membranes were blocked, primary antibodies were incubated overnight at RT, followed by secondary antibodies, and incubated for 2 h at room temperature. As primary antibodies, rabbit monoclonal anti-phospho-NF-κB p65 (ser536) (1:1000) (Cell Signaling, Danvers, MA, USA), rabbit monoclonal anti-phospho-MAPK p38 (Thr180/Tyr182) (1:1000), rabbit polyclonal anti-MAPK p38 (1:1000) (Cell Signaling, Danvers, MA, USA), rabbit polyclonal anti-NF-κB p65 (C-20) (1:1000) (Santa Cruz Biotechnology, Heidelberg, Germany), rabbit polyclonal anti-iκB-α (C-21) (1:1000) (Santa Cruz Biotechnology, Heidelberg, Germany), and rabbit polyclonal anti-actin (I-19) (1:1000) as loading control (Santa Cruz Biotechnology), were used. As a secondary antibody, goat anti-rabbit IgG HRP-conjugated (Millipore) was used. PhemtoMax Super Sensitive Chemiluminescence HRP (Rockland) was used as a substrate and signal acquired on a ChemiDoc XRS (Quantity One software v.4.6.6) (BioRad, Hercules, CA, USA).

### 2.9. Immunocytochemistry

Coverslips containing cells were fixed with paraformaldehyde 4% (*v*/*v*) for 10 min and permeabilized with 0.1% Triton (*v*/*v*) in PBS for 30 min, at room temperature. Then, coverslips were submerged in blocking mixture [1% (*w*/*v*) BSA, 4% (*v*/*v*) FBS, 0.4% (*v*/*v*) Triton, in PBS] for 30 min, before incubation overnight at 4 °C with rabbit polyclonal anti-NF-κB p65 (C-20) antibody (1:200) (Santa Cruz Biotechnology, Heidelberg, Germany) or rabbit polyclonal anti-NFAT1 antibody (1:200) (Cell Signaling, Danvers). Therefore, coverslips were washed with PBS and incubated for 2 h at RT with goat anti-rabbit Alexa Fluor 594 (1:500). Widefield fluorescence images were acquired on a DMRA2 upright microscope (Leica), equipped with a CoolSNAP HQ CCD camera, using the 63× 1.4NA oil immersion objective. Confocal images were acquired on a Leica SP5 live upright confocal (Leica, Wetzlar, Germany), using a 63× 1.3NA oil immersion objective, with spectral detection adjusted for Alexa Fluor 594 and DAPI emissions, using hybrid detectors (HyD). Post-acquiring treatment was performed using ImageJ software version 1.52p (NIH, Bethesda, MD, USA).

### 2.10. Statistics

Data are presented as mean values ± standard errors (SEM) representative of at least three biological replicates. Statistical comparisons were performed using unpaired one-way ANOVA with Tukey post hoc test and considered significant whenever *p* < 0.05. Principal Components Analysis (PCA) on the phytochemical composition of raspberry samples was carried out in Genstat software 20th edition: version 20.1.23942 (VSN International, Hemel Hempstead, UK).

## 3. Results

### 3.1. (Poly)phenol Profiles and Effects of In Vitro Digestion

The (poly)phenol profiles of the 15 raspberry cultivars and genotypes showed considerable diversity in anthocyanins, flavonols, ellagitannins, and ellagic acid derivatives (Appendix A). Such genotypic-derived variation has been noted previously [21,26]. Using PCA, the first two PC scores represented ~50% of variation in the data and illustrated the phenolic variation between samples (Appendix A). From this data, two raspberry cultivars (Glen Ericht and the industry standard, Tulameen) and three genotypes (00123A7, 0304F6 and 2J19) were selected as having the most diverse and distinct compositions. Glen Ericht and Tulameen had the highest anthocyanin contents, whereas 2J19, a yellow raspberry, had an extremely low anthocyanin content (Appendix A). 00123A7 had the highest total flavonol content and 0304F6 the lowest (~8-fold less). Other than 2J19, 00123A7 had the lowest anthocyanin-to-flavonol ratio (~15:1), whereas Glen Ericht had the highest (~88:1). There was less variation in total ellagic acid derivatives and ellagitannin levels between these raspberries (Appendix A).

The phenolic diversity of GIB fractions after IVD was also assessed (Appendix A) and their separation by PCA illustrated that these had retained their diversity (Appendix A). After IVD, the GIB fractions from the red raspberries were generally depleted in anthocyanins over the original extracts with major anthocyanins (e.g., cyanidin sophoroside and cyanidin sophoroside rhamnoside), reaching ~10% recovery but with certain anthocyanins absent (e.g., pelargonidin rutinoside; Appendix A). The poor recovery of anthocyanins is predictable as the pancreatic conditions, i.e., pH > 7, availability of O_2_ and possible binding by pancreatic components, can all contribute to losses [27]. The flavonols also showed low recovery (between 4–15%), but there was no apparent link between stability and their different structures (aglycone or sugar decoration).

The ellagic acid derivatives present in the GIB samples showed recoveries averaging ~15%. Notably, ellagic acid (EA) reached ~25% recovery across the varieties, possibly due to its formation as a degradation product of ellagitannin breakdown [28]. Predictably, the acetylated EA derivatives were less stable (average 7% recovery) whereas methyl ellagic acid derivatives reached ~25% recovery.

The main ellagitannin components were also greatly affected. Lambertianin C had a maximum recovery of ~ 0.5% with Sanguiin H6 at ~ 4%. The higher recovery of Sanguiin H6 might relate to its formation as a degradation product of Lambertianin C under the slightly alkaline IVD conditions [24]. Indeed, ellagitannin degradation products were apparent in the GIB fractions (Appendix A), albeit in small amounts, e.g., the component equivalent to Lambertianin C minus one EA group only amounted to ~ 0.2% of the original Lambertianin C content but it was more abundant (~1.5 fold higher) than Lambertianin C in the GIB fractions. The other ET degradation products showed recoveries ranging from 0.04 to 1.7% of Sanguiin H6 in the original sample but all were less abundant than Sanguiin H6 in the GIB fractions. Various phenolic acids were also present in the GIB fractions including protocatechuic acid and hydroxybenzoic acid. Although possibly indicative of breakdown of anthocyanins, their levels in the GIB fractions were very low, generally 10-fold less than ellagic acid.

The relative composition of the GIB fractions (Figure 1) showed that all the red raspberries were similar. Although the total amounts of anthocyanins were much reduced compared to the original raspberries, they were still the major components (52 to 72% total) in relative terms, followed by ellagitannins and ET degradation products. The yellow raspberry, 2J19, had a very distinct and different GIB composition as it effectively lacked anthocyanins and was enriched mainly in ellagitannins, ET degradation products and EA derivatives. As the GIB fractions were applied in equal amounts of total phenols, these relative compositions describe the components that could be responsible for effects on the cell lines.

### 3.2. GIB Fractions Were Not Cytotoxic at Physiological Concentrations

Each of the GIB fractions was tested for cytotoxicity at physiologically relevant concentrations described for (poly)phenols in human circulation (i.e., 0–2 µg mL^−1^) [29]. The concentrations caused no significant microglial cytotoxicity, even at incubation periods of 24 h (Appendix A). Also, the incubation period used in this cytotoxicity assay, 2–6 h, exceeds the circulation period of (poly)phenol metabolites normally found in the blood stream after an acute ingestion of raspberries [30]. Therefore, we choose to use 1 µg GAE·mL^−1^ as the maximum dose to assess the anti-inflammatory capacity of the GIB fractions, and also as a representative non-cytotoxic physiological concentration. In further experiments, pre-treatments of 6 h were chosen as representative of likely circulation time and because 0–6 h had already been used in our previous study of a GIB fraction [10].

### 3.3. GIB Fractions Richer in ET and Derivatives Inhibited Microglial Pro-Inflammatory Activation

The five GIB fractions (Glen Ericht, 0304F6, 00123A7, Tulameen, and 2J19) were investigated for their potential to attenuate the expression of microglia inflammatory markers TNF-α and CD40, as well as ROS, NO and superoxide (O_2_^🞄−^). Microglia were subjected to 6 h pre-treatment with the GIB fractions, followed by 24 h of LPS stimulation, before analysis. Pre-treatments with every GIB fraction significantly suppressed the secretion of the immune modulator NO by LPS-stimulated microglia compared to LPS-stimulated control cells (Figure 2A). All raspberry GIB fractions demonstrated equally high potential for NO scavenging, which might also suggest a reduction in microglial iNOS activity and/or expression. Differences in the anti-inflammatory potential of the GIB fractions were observed for TNF-α, with Glen Ericht, 0304F6, and 2J19 being the most effective in reducing this cytokine release (Figure 2B). A similar pattern was observed for the quantification of CD40 expression in microglia (Figure 2C) with a small but significant attenuation in cells expressing CD40^high^ also confirmed after pre-treatment with the Tulameen GIB fraction. Although the levels of intracellular superoxide, a typical ROS generated by activated microglia, tended to be reduced by the Glen Ericht, 0303F6 and 2J19 GIB fractions (Figure 2D), only the 2J19 GIB fraction showed significant attenuation (*p* < 0.01).

Together, these results demonstrate that the different constitutions of the five raspberry GIB fractions differently affected the inflammatory response of LPS-stimulated microglia. The NO suppressive effect aside, the 00123A7 GIB fraction caused the least anti-inflammatory activity, with no significant differences against LPS-stimulated control for TNF-α, CD40 and intracellular superoxide accumulation. Conversely, 2J19 exhibited consistent anti-inflammatory potential for all pro-inflammatory markers. Moreover, 2J19 repressed each marker more effectively than other GIB fractions (e.g. reductions of about 70% in NO, 20% in TNF-α, 44% in CD40 and 60% in superoxide). According to their phenolic profiles (Appendix A and Figure 1), the GIB fractions from 00123A7 and 2J19 differed mainly in that 2J19 effectively lacked anthocyanins but also had much higher levels of ETs, ET degradation products and EA derivatives. Interestingly, the GIBs from Glen Ericht and 0304F6, which were the next most effective (both causing significant reductions in CD40 and TNF-α release), had the next highest levels of ETs, ET degradation products and EA derivatives.

### 3.4. J19 GIB Fraction Induced the Release of IL-10, an Important Microglial Anti-Inflammatory Marker

IL-10 is frequently described as a resolution-oriented, anti-inflammatory cytokine that inhibits inflammasome activation in microglia [31]. Therefore, we quantified IL-10 release in LPS-stimulated microglia pre-treated either with 2J19 or 00123A7 GIB fractions, as they caused the most contrasting anti-inflammatory effects (Figure 2E). Although LPS-stimulation *per se* did not significantly alter IL-10 release comparing to control treatments, pre-incubation with 2J19 significantly prompted IL-10 release in LPS-stimulated N9 microglia. Conversely, 00123A7 did not exhibit any significant effect. Previous studies have already provided evidence that certain (poly)phenols are able to induce IL-10 release, as some found in red grapes [32] or green tea [33], but this is the first report of raspberry (poly)phenols exhibiting such an effect. Thereby, we conclude that the higher levels of ETs, ET degradation products and EA derivatives in the 2J19 GIB sample not only attenuate pro-inflammatory markers, but also are capable of inducing a direct anti-inflammatory action by inducing IL-10 release, hampering microglial pro-inflammatory cascades as the inflammasome (Figure 2E).

Nevertheless, the molecular mechanisms of action behind such anti-inflammatory activity require further exploration. Literature increasingly suggests that (poly)phenols exert beneficial bioactivities by modulating upstream canonical pathways rather than just their downstream molecular mediators as ROS or cytokines [25]. Moreover, few studies examine (poly)phenol metabolites that may arise in vivo after digestion, becoming available in the gut for uptake into circulation and leading to biological effects at target tissues, such as the brain. Therefore, we further tested 2J19 and 00123A7 GIB fractions as modulators of three major upstream canonical pathways related to neuroinflammation (e.g., MAPK, NFAT and NF-κB).

### 3.5. J19 GIB Fraction Inhibited MAPK Signaling by Repressing p38 Phosphorylation at Thr180/Tyr182

The GIB fractions from 2J19 and 00123A7 were examined for their ability to inhibit the MAPK signaling pathway, through analysis of p38 phosphorylation status. Phosphorylation of p38 is mediated by upstream kinase members of MAPK family [34] and is an indicator of the activation of this pathway. Notwithstanding the huge impact of MAPK in the overall inflammatory cascade, this signaling pathway extends its regulatory functions to many other general cellular aspects, such as mitosis, cell survival or apoptosis [34]. Microglia cells were stimulated with LPS for 15, 30 and 60 min to monitor the acute inflammatory response after pre-treatment with 2J19 or 00123A7 GIB fractions. As expected, the extent of phosphorylation on the p38 protein greatly increased after LPS treatment over 60 min, as noted by the phosphorylated p38/total p38 ratio (Figure 3A). Only microglia pre-treated with 2J19 fraction showed a significant reduction of p38 phosphorylation when compared to the control with LPS (Figure 3B). However, this effect was only significant at 30 min, which may indicate that inhibition of p38 phosphorylation by 2J19 treatment is temporary. Conversely, pre-treatment with 00123A7 GIB fraction did not significantly change the phosphorylation ratio of p38 compared to control at any LPS time point (15, 30 and 60 min). Nevertheless, even if temporary, the inhibition of p38 phosphorylation by 2J19 could have a powerful impact on the overall inflammatory cascade, which might be critical in a disease context.

### 3.6. NFATc1 Nuclear Translocation Is Inhibited by 2J19

Although not extensively studied in microglia, NFAT is a key pathway that regulates microglial activation and the release of inflammatory cytokines such as TNF-α [35]. Unlike NF-κB and MAPK, this pathway is preferentially activated upon microglial exposure to endogenous stimuli, rather than to exogenous stimuli as the bacterial endotoxin LPS [35]. In that regard, ATP is the most described inflammatory stimulus for NFAT activation [36]. In neurodegeneration, ATP is released by damaged neurons and it functions as a potent chemoattractant molecule for microglia [36]. NFAT is a calcium-dependent transcription factor that remains phosphorylated in its inactive cytosolic form. Once activated, calcineurin removes the phosphorylated residues, activating NFAT and causing migration into *nuclei*. Microglia express only specific NFAT isoforms, with NFATc1 being one of the most abundant [35]. We analyzed the potential of the GIB fractions to inhibit this pathway, by monitoring the nuclear translocation of NFATc1 (Figure 4).

ATP-stimulated microglia clearly showed nuclear translocation of NFATc1 compared to untreated cells (Figure 4). Notably, pre-treatment with 2J19 GIB fraction highly repressed such ATP-induced nuclear translocation of NFATc1. As with MAPK, pre-treatments with the 00123A7 GIB fraction do not show any effect. The high capacity of 2J19 GIB fraction to repress NFAT activation upon ATP stimulation further suggests that the (poly)phenols enriched in this GIB fraction may potentially counteract neurodegenerative-associated microglia activation.

### 3.7. J19 GIB Fraction Represses LPS-Mediated NF-κB Activation by Reducing p65 Nuclear Translocation and Phosphorylation at Ser536

Once activated, NF-κB is a powerful cellular transcription factor, regulating many signaling cascades and the transcription of several genes involved in cell survival, differentiation and proliferation [37]. Like MAPK, NF-κB extends its transcriptional regulation to the immune response, and as result, NF-κB dysregulation is a common hallmark in several pathologies, including neuroinflammatory disorders. Although it is a classical inflammatory pathway, NF-κB requires a multiple-step analysis to draw precise conclusions about its activation status. In the inactive state, the NF-κB effector proteins (mostly p50 and p65) are present in the cytoplasm, sequestered by inhibitory IκB proteins that mask the nuclear localization signaling [37]. Upon exposure to pro-inflammatory stimulus, such as lipopolysaccharide (LPS), ROS, TNF-α or interleukin-1β, IκB proteins are phosphorylated and degraded, allowing phosphorylation and activation of NF-κB effector proteins and their translocation into nucleus [37]. For these reasons, we assessed both p65 nuclear translocation and NF-κB p65 phosphorylation ratio as well as monitored IκB-α levels by Western blotting. The 2J19 GIB fraction showed a remarkable capacity to inhibit p65 nuclear translocation (Figure 5A) in LPS-stimulated microglia, which was not displayed by the 001213A7 GIB fraction. A similar effect was previously demonstrated for specific (poly)phenol metabolites including pyrogallol-sulfate, a low molecular weight (poly)phenol metabolite that appears in circulation resulting from microbiota catabolism [25].

To determine by which mechanism p65 translocation was inhibited by 2J19, we first monitored the levels of the NF-κB inhibitory protein, IκB-α (Figure 5B,C). We have also previously shown that some circulating (poly)phenol metabolites are capable of inhibiting NF-κB activation by accelerating the recovery of IκB-α protein levels after LPS stimulation [25]. However, neither of the tested GIB fractions (00123A7 or 2J19) showed significant effects on microglial IκB-α protein levels at any of the tested timepoints (Figure 5B,C). Finally, we analyzed the phosphorylation status of p65 NF-κB at serine 536 (Figure 5D,E). Phosphorylation of p65, especially at serine 536, is known as a crucial modification that favors the ability of p65 to translocate into the nucleus and trigger inflammatory gene expression [38]. Indeed, p65 phosphorylation at serine 536 was inhibited by the pre-treatment with the 2J19 GIB fraction with the greatest effect after 30 min of LPS stimulation (Figure 5E). Moreover, 2J19 pre-treatment was capable of delaying p65 phosphorylation from 15 min to 60 min after LPS stimulation (Figure 5D). Once again, the 00123A7 GIB fraction showed no significant effect at any timepoint. This emphasizes that the (poly)phenol metabolites present in the 2J19 fraction can modulate NF-κB signaling by specifically inhibiting p65 phosphorylation and nuclear translocation at low, physiologically relevant concentrations. Specific (poly)phenols such as apigenin, luteolin, genistein, 3′-hydroxy-flavone [39], and epigallocatechin-3-*O*-gallate [40] were previously reported as being capable of inhibiting phosphorylation. However, it is unclear how selective 2J19-mediated phospho-inhibition can be, and the possible off-target effects in other important pathways, cells, organs, systems, and disease contexts.

## 4. Discussion

It is important to highlight the fact that this study employed GIB fractions produced by an in vitro digestion model, which approximates the components available for gastrointestinal uptake, and that this aspect is not frequently considered in other studies. Moreover, the GIB fractions were used at near physiological serum levels (according to literature [29]). This study is the first to compare the anti-neuroinflammatory properties exhibited by GIB fractions obtained from raspberry cultivars/genotypes (breeding lines) with distinct (poly)phenol profiles from similar matrices. Interestingly, 2J19 and 00123A7 were the GIB fractions that most contrasted in their (poly)phenolic constitution and also in their anti-inflammatory activities.

Reduction in NO accumulation aside, the 00123A7 GIB fraction exhibited, by far, the weakest anti-inflammatory potential, especially compared to 2J19. The 00123A7 GIB fraction had the highest levels of flavonols and the lowest levels of ellagic acid conjugates. Nonetheless, this fraction was also rich in anthocyanins, which were negligible in the 2J19 fraction. Besides the low anthocyanin content, 2J19 had highest levels of ellagitannins and EA derivatives. Ellagic acid was previously described as having a beneficial role towards neuroinflammation [41], but only recently have ellagic acid conjugates been demonstrated to exhibit such potent effects [42]. The evaluation of these raspberries, with such contrasting phenolic compositions, allowed us to infer that a higher anthocyanin content *per se* may not generate greater anti-neuroinflammatory activity, despite previous reports that diets richer in flavonoids such as anthocyanins correlated with lower rates of inflammation in human population studies [43]. Besides, anthocyanins are known to be highly susceptible to gastro-intestinal conditions, and bioactivities observed in vitro may not be supported in vivo [19].

As the most anti-inflammatory GIB fraction, 2J19 not only inhibited the pro-inflammatory markers TNF-α and CD40, but also induced the anti-inflammatory cytokine IL-10 in LPS-stimulated microglia (Figure 6). In addition, this GIB fraction exhibited the capacity to repress two important oxygen species (NO and superoxide), which are directly involved in peroxynitrite formation, a microglial product that has long been recognized to mediate LPS and amyloid-β neurotoxicity [44]. Moreover, unlike any other tested GIB fraction, 2J19 inhibited MAPK p38 phosphorylation, NFATc1 nuclear translocation and NF-κB p65 nuclear translocation and phosphorylation (Figure 6). Despite its essential role in immunity and inflammation, the MAPK p38 pathway has the broadest and most general impacts. This pathway has long been targeted in human pathology, either in cancer, cardiovascular dysfunction or neurodegenerative diseases [34]. Thus, the inhibition of MAPK p38 by 2J19 suggests it may also be worthy of investigation for other pathological contexts. The NFATc1 and NF-κB p65 pathways each play a critical role in microglia. While NFATc1 typically is activated in response to endogenous triggers, such as calcium fluctuations [45] or excessive extracellular ATP [36], NF-κB p65 is activated upon exposure to a large array of stimuli, via receptors from large families as Toll-like receptors (TLRs) or tumor necrosis factor receptors (TNFRs), being commonly associated with neuroinflammation and neurodegeneration [37]. However, both pathways regulate and are regulated by common cytokines, such as TNF-α, and converge at important functions such as microglia activation and the overall neuroinflammation process [35,46]. Various studies have reported reduced microglial pro-inflammatory activation by specifically inhibiting NFAT [47] or NF-kB [48], and here we demonstrated that 2J19 (poly)phenol metabolites are capable of similar effects. In addition, the effects were simultaneously observed in different pathways, towards LPS and/or ATP stimulation.

Our data suggests that the anti-inflammatory capacity of the 2J19 GIB fraction essentially relies on the ability to modulate phosphorylation status within specific pathways. For instance, NF-κB and MAPK share many common receptors, such as TLR4 (the canonic LPS receptor), and several receptors of the TNFR family, such as CD40 (Figure 6). Each pathway is regulated by phosphorylation events mediated by upstream kinases (IKK for NF-kB p65 and MAPKK for MAPK p38). Our results suggest that 2J19 could be a source of anti-inflammatory compounds (presumably EA and its derivatives/conjugates) able to inhibit such mechanisms (Figure 6).

A previous study has shown that EA inhibits human platelet activation upon stimulation with hydrogen peroxide, by inhibiting phosphorylation events of MAPK p38, Akt, phospholipase C, and protein kinase C [49]. In addition, EA was reported to inhibit tau phosphorylation, one of the prime suspects involved in AD pathology, leading to learning and memory improvements in a mice model of AD [50]. In a more recent study, EA revealed a strong binding affinity for sphingosine kinase 1, repressing phosphorylation by restricting ATP accessibility [51]. However, the evidence of such phospho-inhibitory potential does not explain the observed repression of NFAT activation, nor the increase in IL-10. Indeed, both the NFAT inactivation and IL-10 release involve a direct or indirect phosphorylation event, in opposition to NF-kB p65 and MAPK p38 which do not involve phosphorylation in inactivated forms (Figure 6). It is possible that other 2J19 constituents (besides EA) are involved in such selective pathway-inhibitory mechanisms. Therefore, the combination of (poly)phenols found in 2J19 GIB fraction appears to have a dichotomous effect, specifically inhibiting or inducing phosphorylation events, depending on the pathway in question. Further studies are required to explore this dichotomy, as well as the possible inhibition of IKK and MAPKK kinases cascades by 2J19 (poly)phenols.

## 5. Conclusions

In toto, this study strongly suggests that anthocyanins may not be the major class of anti-neuroinflammatory compounds in raspberry fruits, in opposition to ellagic acid and its derivatives/conjugates. From the five raspberries selected from the Hutton breeding program, 2J19 was the most distinctive one with almost absent anthocyanin levels and a rich profile of ellagic acid, ellagitannins and EA conjugates. Remarkably, the (poly)phenol constitution of 2J19 GIB fraction not only inhibited microglial pro-inflammatory activation, but also stimulated the release of the anti-inflammatory cytokine IL-10 under pro-inflammatory conditions. Moreover, we demonstrated that 2J19 polyphenols repress the activation of canonical inflammatory pathways such as MAPK, NFAT and NF-kB, thus suggesting potential benefits for other inflammatory-associated disfunctions.

Therefore, this study supports that ingestion of raspberries enriched in ellagitannins and ellagic acid derivatives, as identified in 2J19, may have beneficial effects for brain health particularly upon neuroinflammation. This knowledge can be applied practically in several ways for dietary means, drug development, not least of which would be the inclusion of these bioactive compounds as targets in breeding programs. Indeed, the compounds discounted here, anthocyanins, are already embedded breeding targets in many fruit breeding programs, e.g., James Hutton Limited (https://www.huttonltd.com/services/plant-varieties-breeding-licensing/raspberry) for colour, but the emergence of metabolomic and genomic technologies justifies that ellagitannins could become viable targets for elevation through molecular breeding approaches. In addition, identification of the exact (poly)phenol metabolites responsible for the observed anti-inflammatory effects of this fraction could, following categoric characterization of their chemical structures, be used to develop novel enriched foods, nutraceutical formulations for their controlled delivery or even form a chemical skeleton upon which new therapeutic pharma agents can be designed.

## Figures and Tables

**Figure 1 antioxidants-09-00970-f001:**
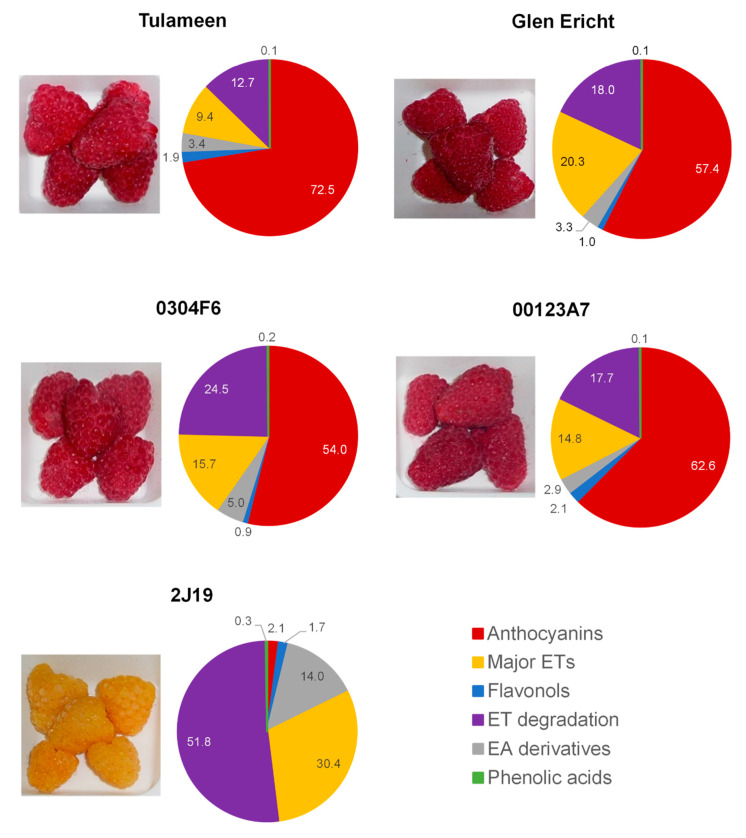
Relative phenolic composition of raspberry GIB fractions. Raspberry cultivars Glen Ericht and Tulameen and genotypes 2J19, 00123A7 and 0304F6 were submitted to in vitro digestion and the recovery of phenolics and identification on degradation products after digestion was achieved using LC-PDA-MS analysis. Values are percentages of total phenolic content, and detailed composition of the GIB fractions is presented in Appendix A. EA—Ellagic acid, ET—Ellagitannins.

**Figure 2 antioxidants-09-00970-f002:**
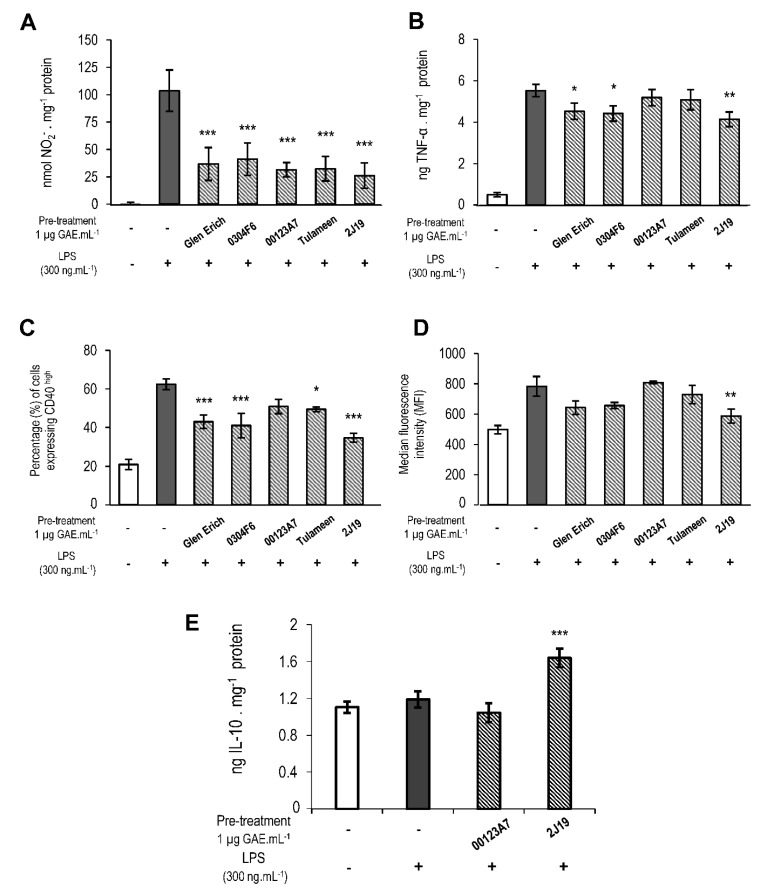
GIB fractions modulate microglial pro- and anti-inflammatory markers, as well as NO and superoxide. N9 microglial cells were pre-treated with each raspberry GIB fraction (1 μg GAE·mL^−1^) for 6 h before stimulation with LPS for 24 h. (**A**) Quantification of the NO release; (**B**) quantification of TNF-α release. (**C**) Percentage of cells that express CD40^high^. (**D**) Median fluorescence intensity (MFI) of dihydroethidium (DHE) as indicator of intracellular superoxide accumulation. (**E**) Quantification of IL-10 release (only 00123A7 and 2J19 GIB fractions were analyzed). Results were normalized for the total protein in each treatment (mean ± SEM; *n* = 3). Solid white bars represent untreated cells (naive); solid grey bars represent LPS-stimulated cells with no GIB fraction pre-treatment (control); striped bars represent the cells pre-treated with the indicated GIB fraction before LPS stimulation. Statistical comparisons are respective to the differences between the pre-treatment with each of the GIB fractions (striped bars) and the control (solid grey bars), and denoted as *** *p* < 0.001; ** *p* < 0.01 and * *p* < 0.05.

**Figure 3 antioxidants-09-00970-f003:**
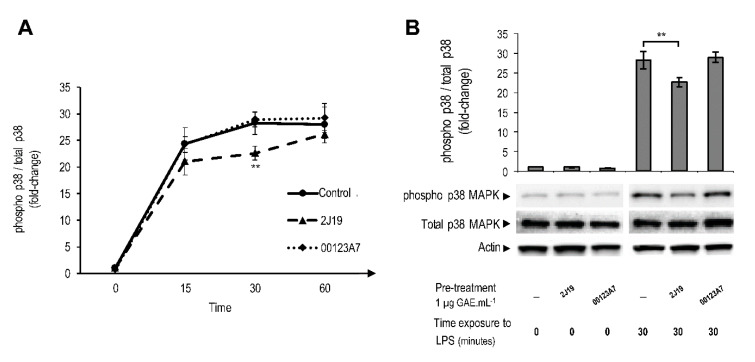
Modulation of MAPK p38 phosphorylation upon pre-treatment with 00123A7 and 2J19 GIB fractions. N9 microglial cells were pre-treated either with 2J19 or 00123A7 GIB fractions 6 h before LPS stimulation. (**A**) p38 phosphorylation (Thr180/Tyr182) ratio kinetics during 0, 15, 30, and 60 min of LPS stimulation. LPS-stimulated (control) cells are represented as 
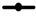
; cells treated with 2J19 prior to LPS stimulation are denoted as 
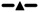
 and cells treated with 00123A7 prior to LPS stimulation are denoted as 
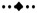
. (**B**) Comparison of the p38 phosphorylation (Thr180/Tyr182) ratios between each type of pre-treatment at 0 and 30 min of LPS stimulation (mean ± SEM; *n* = 3). Statistical differences between 2J19 pre-treated cells and the LPS-stimulated control cells are denoted as ** *p* < 0.01.

**Figure 4 antioxidants-09-00970-f004:**
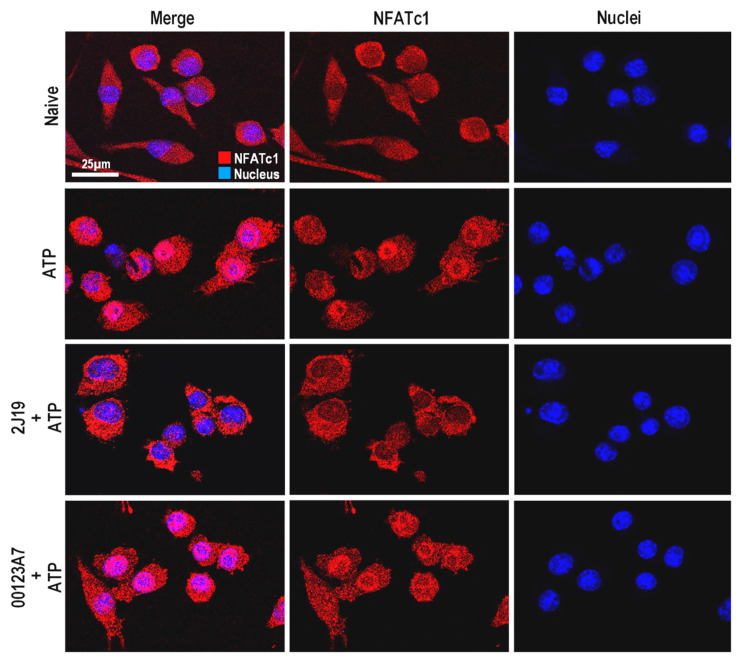
Modulation of NFATc1 nuclear translocation by 00123A7 and 2J19 GIB fractions. N9 microglial cells were pre-treated with 2J19 or 00123A7 GIB fractions for 6 h before ATP (3mM) stimulation for 60 min. Confocal immunofluorescence images of NFATc1 (red) and nuclei (blue) stained with DAPI. Each capture is representative of, at least, three independent biological replicates.

**Figure 5 antioxidants-09-00970-f005:**
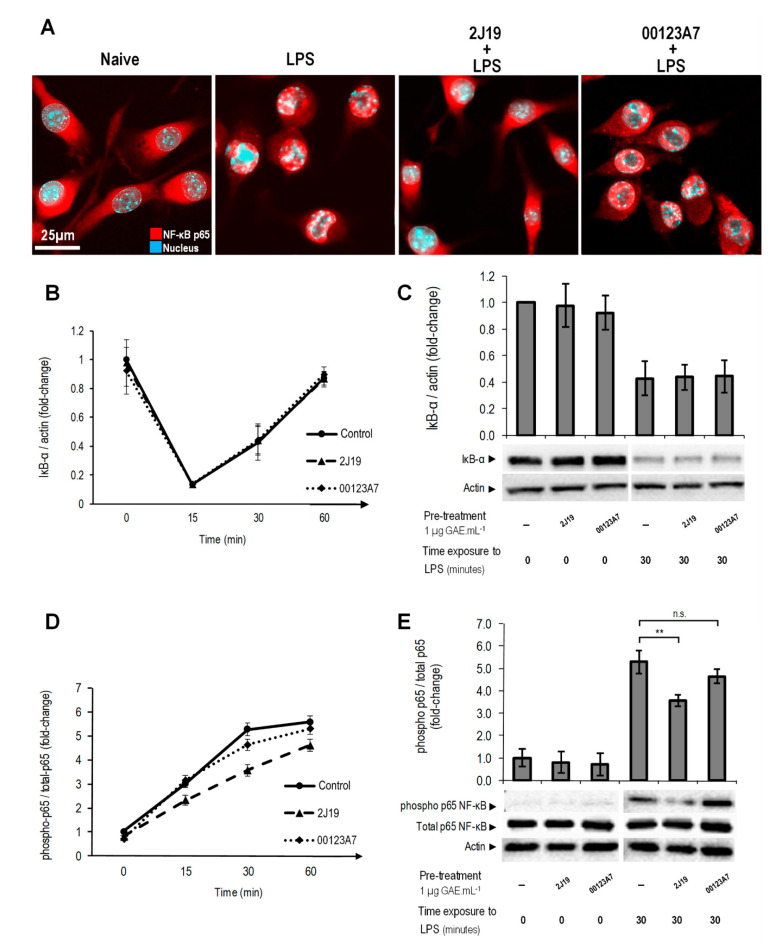
Modulation of NF-kB pathway activation by 2J19 and 00123A7 GIB fractions. N9 microglial cells were incubated with 2J19 or 00123A7 for 6 h before LPS-stimulation. (**A**) Immunofluorescence images for the evaluation of NF-κB p65 (red) translocation into the nuclei (cyan). Each capture was chosen as representative of, at least, three independent experiments. (**B**) Kinetics of IκB-α protein levels and (**D**) NF-κB p65 phosphorylation (Ser536) status, both after 0, 15, 30, and 60 min of LPS stimulation. LPS-stimulated cells (control) are denoted by 
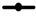
; cells treated with 2J19 prior to LPS stimulation are denoted by 
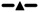
, and cells treated with 00123A7 prior to LPS stimulation are denoted by 
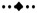
. (**C**) IκB-α protein levels and (**E**) p65 phosphorylation (Ser536) ratios for each GIB fraction at 0 and 30 min of LPS stimulation (mean ± SEM; *n* = 3). Statistical differences *vs* respective timepoint control are denoted as ** *p* < 0.01; “n.s.” means not statistically different.

**Figure 6 antioxidants-09-00970-f006:**
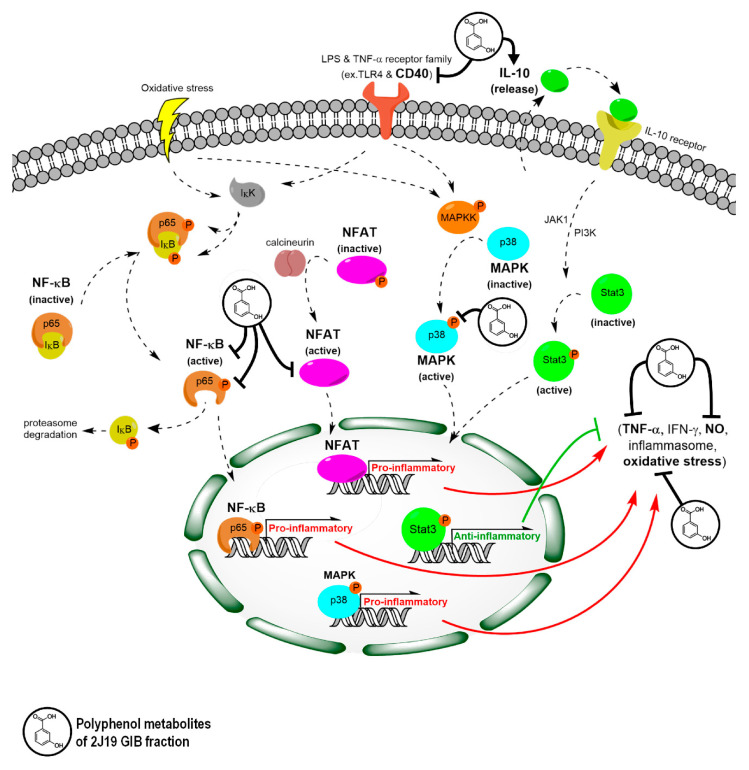
Scheme of N9 microglia NF-kB, NFAT, MAPK, and IL-10/STAT3 inflammatory pathways modulated by 2J19 GIB fraction. Pre-treatment with 2J19 GIB fraction attenuates N9 microglial pro-inflammatory response to LPS by repressing the NF-κB pathway via inhibition of both p65 phosphorylation and nuclear translocation. MAPK was also inhibited via p38 phosphorylation, as well as NFAT via nuclear translocation upon ATP stimulus. The inhibition of these pathways not only led to a reduction in pro-inflammatory markers CD40, TNF-α, NO, and superoxide, but also led to an increased release of the anti-inflammatory cytokine IL-10. Solid black lines represent evidence-supported effects of 2J19 GIB (poly)phenols; dashed black arrows indicate pathway activation steps supported by literature; solid red arrows represent pro-inflammatory gene expression leading to the indicated molecules; a solid green line indicates the effects of anti-inflammatory gene expression that inhibits the indicated molecules.

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
