# Peer review of "Bioaccessible Raspberry Extracts Enriched in Ellagitannins and Ellagic Acid Derivatives Have Anti-Neuroinflammatory Properties"

_antioxidants, 2020, doi:10.3390/antiox9100970_

Round 1

Reviewer 1 Report

Dear Authors

In this study, authors used five different raspberry genotypes which were markedly different in their (poly)phenol profiles to assess the relation between (poly)phenol classes and bioactivity. They also conducted the gastro-intestinal bioaccessible fractions, and evaluated for anti-inflammatory potential using LPS-stimulated microglia. Results indicated that the fraction enriched in ellagitannins, their degradation products and ellagic acid, attenuated pro-inflammatory markers and mediators CD40, NO, TNF-α and intracellular superoxide via NF-κB, MAPK and NFAT pathways. Surprisingly, the anthocyanins, which are embedded breeding targets in many fruit breeding programs for colour, but the emergence of metabolomic and genomic technologies justifies that ellagitannins could become viable targets for elevation through molecular breeding approaches. Further studies on identification of the exact (poly)phenol metabolites responsible for the observed anti-inflammatory effects, following categoric characterization of their chemical structures, can be used to develop novel enriched foods, nutraceutical formulations for new therapeutic pharma agents. We look forward to it!

Overall, this manuscript is well organized and comprehensively described, the results appear sound and the conclusions and discussion is well done. The results may interest to readers and I suggest this manuscript can be accepted in present form.

Author Response

The authors are pleased by your appreciation about this work.

Our most difficult task was to make this integrative study interesting enough, not only for breeders, but also for the nutraceutical and/or drug development readership. We are happy with the result, but even more happy because of the positive feedback that we received from reviewers.

Thanks for your input and support,

Reviewer 2 Report

Few errors to correct, highlighted in yellow in the attached file. Interesting, complete, well written article

Author Response

The authors are pleased by your appreciation and hard work checking all the errors. Your corrections are very much appreciated! 

Our most difficult task was to make this integrative study interesting enough, not only for breeders, but also for the nutraceutical and/or drug development readership. We are happy with the result, but even more happy because of the positive feedback that we received from reviewers. 

Thanks for your input and support,

Reviewer 3 Report

The paper presents an evaluation of the neuroprotective effects of 5 raspberry genotypes in a neuroinflammation model based on LPS78 stimulated microglia.

In my opinion the paper is worth studying and the manuscript contains enough original material.

The work is interesting, well planned and described.

The experimental tests are carried out correctly using appropriate methods.

The results are interesting and well statistically analyzed.

Minor corrections:

The quality of the figures is very poor.

Text formatting should be carefully checked.

The language should be modified carefully.

Author Response

The authors are very pleased for your appreciation of this work and for your critical review it. It was our objective to make it interesting enough not only for breeders, but also for the nutraceutical and/or drug development readership.

About your minor corrections:

The quality of the figures was not great on the manuscript but we uploaded the images at high resolution to the platform. In addition, the next version of the manuscript already includes those high resolution images.

We checked up the text formatting issues and changed some language in critical parts.

Once again,

Thanks for your input